

# An advanced tool integrating failure and sensitivity analysis to novel modeling for stormwater flooding volume

Francesco Fatone[1], Bartosz Szeląg[2], Przemysław Kowal[3], Arthur McGarity[4], Adam Kiczko[5], Grzegorz Wałek[6], Ewa Wojciechowska[3], Michał Stachura[7], Nicolas Caradot[8]

[1] Department of Science and Engineering of Materials, Environment and Urban Planning-SIMAU, Polytechnic University of Marche Ancona, 60121 Ancona, Italy
[2] Faculty of Environmental, Geomatic and Energy Engineering, Kielce University of Technology, 25-314 Kielce, Poland
[3] Faculty of Civil and Environmental Engineering, Gdansk University of Technology, 80-233, Gdansk, Poland
[4] Department of Engineering, Swarthmore College, 500 College Ave., Swarthmore, PA, 19081, United States
[5] Institute of Environmental Engineering, Warsaw University of Life Sciences-SGGW, 02-797 Warsaw, Poland
[6] Institute of Geography and Environmental Sciences, Jan Kochanowski University in Kielce, 25 – 406, Kielce, Poland
[7] Faculty of Law and Social Sciences, Jan Kochanowski University, 25 – 406, Kielce, Poland
[8] Berlin Competence for Water, Cicerostr. 24, 10709 Berlin, Germany

*Correspondence to*: Bartosz Szeląg (bszelag@tu.kielce.pl)

**Abstract.** An innovative tool for modelling specific flood volume was presented, which can be applied to assess the need for stormwater network modernisation as well as for advanced flood risk assessment. Field measurements for a catchment area in Kielce, Poland were used to apply the model and demonstrate its usefulness. This model extends the capabilities of recently developed statistical and/or machine learning hydrodynamic models developed from multiple runs of the U.S. EPA's Storm Water Management Model (SWMM) model. The extensions enable inclusion of: 1) characteristics of the catchment, and its stormwater network, calibrated model parameters expressing catchment retention and the capacity of the sewer system, (2) extended sensitivity analysis and (3) risk analysis. Sensitivity coefficients of calibrated model parameters include correction coefficients for percentage area, flow path, depth of storage, impervious area, Manning roughness coefficients for impervious areas, and Manning roughness coefficients for sewer channels. Sensitivity coefficients were determined with regard to rainfall intensity and characteristics of the catchment and stormwater network. Extended sensitivity analysis enabled an evaluation of the variability of the specific flood volume and sensitivity coefficients within a catchment, in order to identify the most vulnerable areas threatened by flooding, Thus, the model can be used to identify areas particularly susceptible to stormwater network failure and the sections of the network where corrective actions should be taken to reduce the probability of system failure. The developed simulator to determine a specific flood volume represents an alternative approach to the SWMM model that, unlike current approaches, is calibratable with limited topological data availability, therefore generates a lower cost due to the less amount and specificity of data required.





**Highlight**

- simulator of a specific volume of flood as an alternative to the SWMM model,

- sensitivity analysis extension considering rainfall and catchment topological data,

- the probability of failure of the stormwater system as a criterion for corrective actions under conditions of uncertainty

## 1 Introduction

Climate change and urbanization are the main factors increasing the pressure on the functioning of sewer networks, in particular components responsible for stormwater management (Miller et al., 2014; Hettiarachchi, et al, 2018; Khan et al, 2022). This results in an increase in the frequency and volume of stormwater flooding, deterioration of the living standards of the inhabitants, and pipes abrasion (Jiang et al., 2018; Zhou et al. 2018; Chang et al. 2020). The literature data (Siekmann et al. 2011) shows that the basis for making decisions on corrective actions (replacement of a pipe, removal of sediments, construction of a reservoir, etc.) is the specific flood volume expressing the volume of stormwater flooding on a unit impervious surface. Limiting values for the specific flood volume have been determined by Siekmann and Pinnekamp (2011), based on simulations for urban catchments, as the basis for the maintenance of the sewage network and the criterion for making decisions on modernization or corrective actions.

In order to obtain a required hydraulic efficiencies, simulation models are typically used to plan corrective actions (Kirshen et al. 2014). For this purpose, mechanistic models are used, such as the USEPA's Storm Water Management Model (SWMM), which account for surface runoff, drainage of the sewage network, and flooding of stormwater during system overload (Guo et al. 2021; Li et al. 2022; Yang et al., 2022). As in the case with other mechanistic models (MOUSE, PCSWMM, MIKE URBAN etc.), SWMM can incorporate the spatial characteristics of a sewage network, as well hydraulic conditions, in calculations that predict and characterize stormwater flooding (Martins et al. 2018; Yang et al., 2020; Ma et al., 2022). However, such models are characterized by high specificity (one model can be used for one catchment), and they require the collection of detailed data and measurements (rainfall, runoff), which is labour-intensive and generates high costs. Moreover, there are a strong interactions between the calibrated parameters (Wu et al. 2013; Chen et al. 2018; Sonavane et al. 2020; Shrestha et al., 2022), leading to uncertainty of simulation results (Ball 2020; Kobarfard et al. 2022; Sun et al. 2022) which complicates to select specified corrective action (Kim et al. 2017; Bobovic et al. 2018; Hung and Hobbs 2018). To solve this problem, an important step in the development of the computational algorithm is the implementation of sensitivity analysis (Fraga et. al. 2016; Cristiano et al. 2019; Razavi and Gupta 2019). Simulations by Szeląg et al. (2021) have shown the influence of uncertainty in calibrated SWMM parameters on the calculation of specific flood volume and degree of flooding, which was also confirmed by the simulations of Fraga et al. (2016) and Kelleher et al. (2017).

To overcome the limitations of MCM, the implementation of statistical and/or machine learning methods seems is a prospective alternative (Rosenzweig et al. 2021; Lei et al. 2021; Bui et al. 2019; Shafizadeh-Moghadam et al. 2018; Chen et al. 2019; Fong and Chui, 2020). ML methods can estimate of specific stormwater flood volume for a catchment area with different topology. However, so far, no simulator model based on statistical and/or machine learning has been developed to





simulate specific stormwater flood volume while taking into account the factors included in mechanistic models (Mignot et
al., 2019; Guo et al. 2021; Rosenzweig et al. 2021). Some progress in application of machine learning methods to simulation
of stormwater flooding has been made. Thorndahl et al. (2008), based on simulation results of flooding from manholes,
including uncertainty of calibrated parameters, developed a model using the FORM (first order reliability model) method. Jato-
Espino et al. (2018) and Li and Willems (2020), conducting simulations with mechanistic models, present models (logisitc
regression) for identification of flooding from a single manhole based on rainfall frequency, catchment and stormwater network
characteristics. Therefore, Szeląg et al. (2022a, 2022b) proposed a models for calculating estimates of stormwater flooding in
a catchment, but due to insufficient data in constructing the model, application is limited. In the aforementioned models,
interactions between land use, catchment and stormwater network characteristics, as well as system capacity were neglected.
However, by omitting these factors, at the spatial planning stage, reduces the applicability of the model.
Another important indicator of proper sewage network management is the assessment of the risk of system failure
(exceed the maximum specific flood volume). Reliable risk assessment requires the integration of mechanistic models,
statistical approach and simulators of rainfall data (Fu et al. 2012; Zhou et al. 2019; Venvik et al. 2020). Most of the methods
(Ursino 2014; Cea and Costabile 2022; Taromideh et al. 2022) focus on determining the impact of climatic changes in rainfall
on the efficiency of the sewage system and include the impact of parameters expressing terrain and sewer retention. Currently,
there is no effective method of risk analysis taking into account the uncertainty of the calibrated parameters to simulate a
specific flood volume for the different urban catchments.
The aim of the article was to develop an innovated simulator, combined with risk assessment and sensitivity analyses
for calculating the specific flood volume, taking into account rainfall data, catchment characteristics and topology. Recognition
of the above factors enabled the application of the proposed logistic regression model to identify stormwater flooding in
catchments with different characteristics, as an alternative approach to the SWMM model. An important aspect of the proposed
approach was the risk assessment of system failure (specific volume of flood exceed 13 $m^3 \cdot ha^{-1}$) and sewage system operation
under uncertainty. Moreover, the methodology presented in the work, integrated with the stormwater flooding simulator,
enabled the identification of the impact of calibrated SWMM parameters on the results of the sensitivity analysis in catchments
with different characteristics. This feature enables building a mechanistic model, which allows appropriate selection of
techniques for measuring input data, which can ultimately reduce the costs of applying the model. The developed methodology
enables the appropriate selection of devices for measuring the flow rate, and their location in the sewage network in the context
of calibrating the catchment model and reducing the costs of flow measurements.

**2 Case study**
The analysed urban catchment is located in the south-eastern part of Kielce, central Poland, Świętokrzyskie region
(Fig.1). Residential districts, public buildings, main and side streets are located in the study area. The catchment area covers
ha and consists of 40% impervious and 60% permeable areas. The road density is 108 $m \cdot ha^{-1}$ (Wałek, 2019), and the terrain
denivelation is 11.20m (the ordinates of the highest and the lowest points of the terrain are 271.20 m and 260 m above sea





level, respectively). The length of the main interceptor channel in the stormwater network is 1569 m, with an average slope of
0.71%. The diameter of the main interceptor expands from 600 to 1250 mm, while the diameters of side sewers vary between
300 and 1000 mm. The slope of the sewers varies between 0.04 and 3.90%.

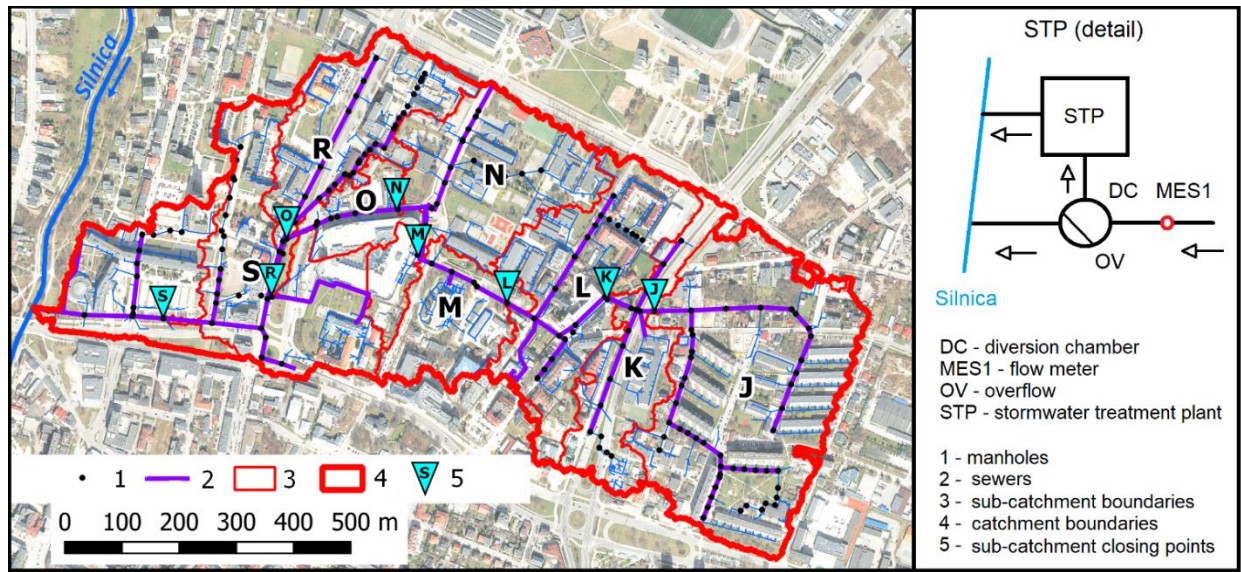


**Figure. 1. Study catchment area (Wałek, 2019).**

The analysed stormwater system is separated from the municipal sewage. Stormwater flows to the division chamber (DC), and
after reaching a depth of 0.42 m it flows into a stormwater treatment plant (STP). During heavy rainfall, when the stormwater
level in the DC exceeds the overflow level (OV), it is discharged by the storm overflow (OV) into the S1 channel, which
transports the stormwater directly to the Silnica river (without treatment). At a 3.0 m distance from the inlet of the main
interceptor to the DC, the flow meter MES1 is installed, which measures the flow rates during heavy rainfall with resolution
of 1 minute. Analysis of data from 2010–2020 showed that during dry periods the measured flow rates varied between 1–9
$dm^3 \cdot s^{-1}$, which indicates that infiltration occurs in the stormwater network. Measurements of stormwater network operation
carried out in the years 2008–2019 indicated that stormwater flooding occurs in the analysed catchment. Taking into account,
159 episodes of rainfall – runoff, within four catchments, 23 cases of flooding were observed. At a distance of 2.5 km from
the catchment boundary, a rainfall measurement station is located, which provides constant measurement of rainfall, with a 1-
minute temporal resolution.

*Sub-catchment division and characteristics*
The analysed catchment was divided into sub-catchments (Szeląg et al. 2022), which constituted study areas for
identification of stormwater flooding. Due to limited amount range of rainfall data, the obtained model for simulation of
stormwater overflow did not include all important factors, such as dry period duration between rainfall events, retention





catchment that impact flooding phenomenon, which meant that the model had limited predictive capability. Detailed
description and justification of sub-catchments used for construction of flooding identification model was presented by Szeląg
et al. (2022). In reference to approach proposed by Duncan et al. (2011), Jato – Espino et al. (2018), Li and Willems (2022),
in the current analysis the number of sub-catchments used for development of a logit model was increased to 8 (Figure 2). The
sub-catchments boundaries together with data on spatial development and stormwater network (Table 1) were determined
based on maps for design purposes, which was discussed in detail by Szeląg (2013).

**Table. 1. Characteristics of sub-catchments**

| No. | F | Imp | Vk | Gk | R.t. | Vkp | dH1 | dHp | Lk | Jkp | Hst | Impd | Gkd | Vrd | Vkd |
|-----|-----|------|--------|---------|------|------|------|------|------|--------|------|------|---------|--------|--------|
|  | ha | - | m³ | m·ha⁻¹ | m | m³ | m | m | m | - | m | - | m·ha⁻¹ | m³ | m³ |
| J | 12.66 | 0.37 | 157.0 | 0.0079 | 1.74 | 33.2 | 0.24 | 0.25 | 96.5 | 0.0036 | 1.42 | 0.40 | 0.0072 | 2159.4 | 2577.2 |
| K | 18.92 | 0.38 | 360.4 | 0.0084 | 1.69 | 28.4 | 0.31 | 1.05 | 56.5 | 0.0055 | 2.36 | 0.40 | 0.0063 | 1886.8 | 2373.7 |
| L | 27.15 | 0.36 | 557.4 | 0.0074 | 2.74 | 29.6 | 0.34 | 1.75 | 59.0 | 0.0058 | 2.36 | 0.42 | 0.0053 | 1496.0 | 2176.7 |
| M | 29.78 | 0.36 | 678.8 | 0.0068 | 4.49 | 48.7 | 0.38 | 1.15 | 62.0 | 0.0061 | 2.32 | 0.43 | 0.0050 | 1373.3 | 2055.3 |
| N | 36.78 | 0.37 | 712.2 | 0.0081 | 4.49 | 48.7 | 0.38 | 1.15 | 62.0 | 0.0061 | 2.32 | 0.44 | 0.0040 | 1061.4 | 2022.0 |
| O | 41.31 | 0.32 | 858.2 | 0.0079 | 5.32 | 16.1 | 0.21 | 1.28 | 20.5 | 0.0102 | 2.31 | 0.49 | 0.0037 | 825.9 | 1876.0 |
| P | 45.42 | 0.37 | 981.9 | 0.0082 | 5.64 | 16.1 | 0.21 | 1.28 | 20.5 | 0.0102 | 2.31 | 0.46 | 0.0027 | 682.2 | 1752.3 |
| R | 48.31 | 0.37 | 981.9 | 0.0088 | 5.64 | 16.1 | 0.21 | 1.28 | 20.5 | 0.0102 | 2.31 | 0.47 | 0.0023 | 553.1 | 1752.3 |
| S | 55.41 | 0.41 | 1240.2 | 0.0092 | 8.47 | 67.5 | 0.67 | 1.8 | 86.0 | 0.0078 | 2.31 | 0.55 | 0.0011 | 258.4 | 1493.9 |

where: F – catchment surface area; Imp – impervious area; Vk – volume of stormwater channel; Gk – length of stormwater
channel per impervious area of the catchment; R.t. – height difference of the channel, Vkp – volume of the channel above the
cross-section of a catchment; dH1 – height difference of the terrain at section above cross-section r; dHp – height difference
at section above cross-section; Lk – length of channel above cross-section of a catchment; Jkp – channel slope above cross-
section of a catchment; Hst – the height of a manhole at cross-section; Imp – impervious area of downstream area; Gkd –
length of a channel per impervious area below cross-section; Vrd – catchment retention above the cross-section calculated as
$Vrd = F \cdot (Imp \cdot d_{imp} + (1-Imp) \cdot d_{per})$, Vkd – total retention of a catchment.

Data were verified using independent analysis performed by Wałek (2019), who used Qgis program to develop spatial
development model and stormwater network for Kielce. Location of closing cross-sections of sub-catchments (J, K, L, M, M,
O, P, R, S) along the main interceptor were additionally supported by simulation results of outflow hydrographs developed by
Wałek (2019) with use of HEC-HSM model as well as by Szeląg et al. (2014, 2022) with use of hydrodynamic model SWMM.

**3 Methodology**
**3.1. Criterion for stormwater system operation and modernisation**



The value of a specific flood volume was defined as stormwater flooding per unit impervious area, which can be
expressed by the following formula (Sinekamp and Pinekamp, 2011):
$$\kappa = \frac{\sum_{i=1}^{K} V_{t(i)}}{A_{imp}} \qquad (1)$$

where: $V_t$ – volume of stormwater flooding from i-th manhole of the stormwater network, $K$ – number of manholes, $A_{imp}$ –
impervious area. Sinekamp and Pinekamp (2011) based on continuous simulations with hydrodynamic models for 3 urban
catchments found that the specific flood volume ranged from 0 - (>20) $m^3 \cdot ha^{-1}$.
On this basis, they established limiting κ values expressing the need to improve the operating conditions of the drainage system.
They showed that for κ > 13 $m^3 \cdot ha^{-1}$ the drainage system requires adaptation This was also confirmed by the calculations of
Kotowski et al. (2014) for the catchment in Wroclaw and Szeląg et al. (2021) for the catchment in Kielce. This allows us to
conclude for urban catchments (Poland, Germany) that the κ value quoted above can be a criterion for making decisions on
corrective actions of the drainage network.
**3.2. Simulator structure and development**
The concept of the proposed of tool based on simulator integrated with the risk assessment and sensitivity analysis to
evaluate operation of sewage system was presented in Fig. 2.Applying the MCM of an urban catchment with separate sub-
catchments (varying land use and topology), a simulator of the specific flood volume was developed as an alternative approach
to the SWMM. A proposed simulator of logistic regression model-based on rainfall data, catchment and stormwater network
characteristics, SWMM parameters (width of runoff path, retention depth of impervious areas, Manning roughness coefficient
of impervious areas, correction coefficient of impervious areas, Manning roughness coefficient of channels). The resulting
tool enables fast analysis of sewer network performance even with limited data access and can be applied to other catchments.
Proposed methodology is based on extension of algorithms given by Szeląg et al. (2021, 2022). In contrast to previous studies
(Szeląg et al. 2022), the current approach took into account the retention of the catchment and the sewer network, and the
performance criterion of the sewer network was the volume of flooding and not just the fact that it occurred. Integration of the
simulator with an analytical relationship for sensitivity coefficient calculations for logistic regression allows fast evaluation of
the impact of MCM parameters on flooding for arbitrary catchment characteristics and topological data. In order to provide
more reliable simulation results, the proposed risk assessment took into account the uncertainty of the SWMM parameters and
enabled the optimisation of the operation of the sewer network based on the maximum allowable values of the channel Manning
roughness coefficients.



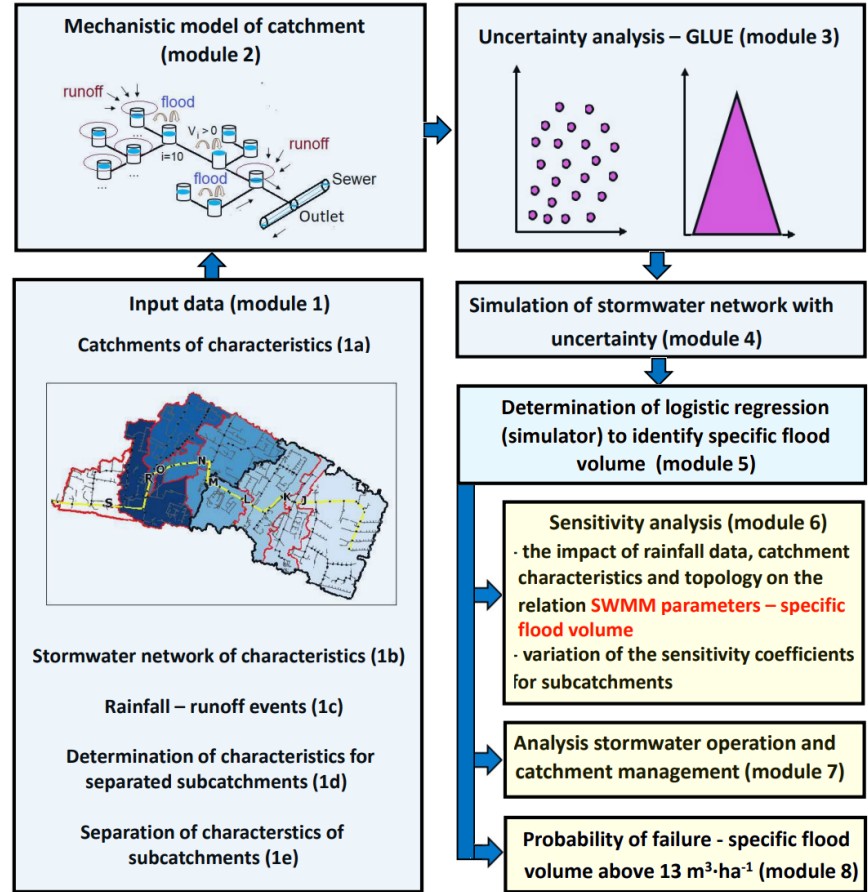


**Figure. 2. Algorithm for developing an advanced tool to simulate a specific flood volume (situation maps in module (1a), (1b) by Walek (2019).**

## 3.3. Algorithm structure

The proposed computation algorithm consists of 9 modules. Modules 1, 2, 3, 4 include identical steps as in the work of Szeląg et al. (2021, 2022). In the present study, the scope of the analyses was extended, as in addition to precipitation data and SWMM parameters (Szeląg et al. 2022), the characteristics of the catchment and the stormwater network of the separated sub-catchments were also included (module 1), which was used to determine the computational model. On the basis of spatial data (1a, 1b), a mechanistic model of the catchment was built (module 2), which allowed to perform an uncertainty analysis using the GLUE method (module 3). On this basis, simulations were performed in separated sub-catchments for rainfall events (1e) under uncertainty (module 4). Based on the simulation results a logistic regression model was developed (module 5) to calculate the local sensitivity coefficients for calibrated SWMM parameters, with regard to rainfall intensity and catchment characteristics (module 6). Modules 1, 2, 3, 4 included analyses to determine a specific flood volume simulator that can be applied to any catchment. Thus, future algorithm implementation for the new catchment, will ultimately include only modules





6, 7, 8. Using adopted rainfall data, the sensitivity coefficients of SWMM model parameters for sub-catchments are computed
and maps showing sensitivity changes in catchment scale are drawn (module 6). While the model is applied to identify
stormwater flooding, the possible methods for improving stormwater network operating are analysed inside module 7, 8.
Computations using the developed algorithm consist of the following steps:
1) collecting of the input data (catchment characteristics – 1a, stormwater network characteristics – 1b, rainfall – runoff
episodes – 1c), separation of independent rainfall episodes – 1d, division and determination of characteristic of sub-catchments
– 1e,
2) development of hydrodynamic model (module 2) based on catchment characteristics (1a) and stormwater network
characteristics (1b),
3) conducting of uncertainty analysis with GLUE method (section 3.3.3) using hydrodynamic model of a catchment based on
rainfall – runoff episodes (1d),
4) using independent rainfall events (1d) simulations with hydrodynamic model including uncertainty of calibrated parameters
according to points (4a, 4b, 4c) are conducted;

a) simulation of SWMM parameters (*a posteriori distribution*) in Table S1 using the results of uncertainty analysis,

b) simulation of stormwater network operation during independent rainfall events (1d) including uncertainty (4a),

c) computation of specific flood volume in each sample of independent rainfall events in sub-catchments;

transformation of determined $\kappa$ values to classification data (section 4a),

5) determination of logistic regression simulator SWMM of specific flood volume as alternative to MCM model based on
results of computations in point 4c,
6) sensitivity analysis:
a) computations of sensitivity coefficients (with regard to SWMM parameters) for assumed rainfall data and catchment
characteristics,
b) computations of sensitivity coefficients for sub-catchments (J, K, L, M, N, O, P, R, S),
7) application of developed logistic regression model for amelioration of stormwater network operation,
a) analysis of the impact of corrective variants on sensitivity coefficients in sub-catchments,
8) analysis of failures occurrence.

**3.3.1. Determination of independent rainfall events (module 1e)**

Determination of independent rainfall events for the period 2010 - 2021 was based upon criteria defined in DWA A-

118 (2006) guidelines. The minimum time period between independent rainfall events was set as 4.0 hours. Computation of
stormwater flooding was performed for rainfall events with a minimum depth of $P_t = 5.0$ mm (Fu and Butler, 2014) and only
for those events that resulted from convection rainfalls (i.e., rainfall duration below 120 min). For the analysed catchment, it
was indicated that stormwater flooding occurs for C = 2, 3, 5 and rainfall duration $t_r = 120$ min (Szeląg et al., 2021). The
computed values of specific flood volume (the upper limit of 95% confidence interval) are $\kappa = 45$ m$^3 \cdot$ha$^{-1}$. Analyzing of the





rainfall data, it was observed that the number of rainfall events with depths of $P_t$ = 5.2–42 mm ranged from 12 to 30 in each
year (210 rainfall events altogether), while the rainfall durations were between $t_r$ = 15 –120 min.

**3.3.2. Hydrodynamic catchment model (module 2)**
Stormwater flooding volume calculations were performed with the SWMM model using the „Flooding" function
(Szeląg et al. 2021). Based on the results of $Q(t)$ for j – manholes (j = 1, 2, 3 ..., k) in the sub-catchments (J, K, L, M, N, O,
P, R, S), the total flooding volume $V_j = \int Q(t)dt$ was determined, which allowed specific flood volume (κ) values to be
determined from Equation (1).
The model of analysed catchment covers 62 ha and is divided into 92 sub-catchments with areas varying from 0.12
to 2.10 ha and impervious areas ranging 5 to 95%. The model comprises 82 nodes and 72 sections of channels. At the
calibration stage method of the „trial and error", the mean retention of the catchment equal of 4.60 mm. The Manning
coefficient of impervious areas was found to be 0.025 $m^{-1/3}$·s and 0.10 $m^{-1/3}$·s for pervious areas. The flow path width was
determined using the formula W=α·A$^{0.50}$, where: α = 1.35. Catchment model calibration performed by Szeląg et al. (2021)
indicated that for 6 rainfall-runoff events, a very good fit of modelling outflow hydrographs to measurement results was
obtained (Nash - Sutcliff coefficient was 0.85 - 0.98, coefficient of determination was equal to 0.85 - 0.99, hydrograph volumes
and maximum flows did not exceed 5% compared to measurement data).

**3.3.3. Uncertainty analysis – GLUE (module 3)**
In the GLUE method, the identification of model parameters was considered as a probabilistic task due to the large
number of parameters characterizing processes occurring in urban catchments (runoff, infiltration, flow in stormwater conduits,
flooding) – Szeląg et al. (2021), Kiczko et al. (2018), Mannina et al. (2018). The identification of model parameters in the
GLUE method depends on the transformation of an *a priori distribution* to an *a posteriori distribution* by means of a likelihood
function $L(Q/\theta)$, which determines the probability of a combination of parameters depending on the quality of fit of the
calculation result to the measured values. Uniform distribution of SWMM parameters was assumed (Table S1). Mathematical
models used for description of surface runoff usually do not include runoff distribution and at most they include the range of
admissible values of parameters resulting from their physical interpretation (Dotto et al., 2014; Knighton et al., 2016).
Identification of distributions *a posteriori* and determination of likelihood functions the rainfall - runoff episodes 30 May 2010
and 8 July 2011 were used, while for verification the episodes from 15 September 2010 and 30 July 2010 were applied. Subsequent
computation steps of GLUE analysis were discussed in detail in Supplementary Information (Section 1).

**3.3.4. Simulation of stormwater network operating with regards to uncertainty (module 4)**
Based on the results of GLUE (*a posteriori distribution* SWMM parameters, 5000 sampling), the computation of
stormwater network was performed for separate 175 independent rainfall events and 9 subcatchments; 35 events were used to
validate the model. The values of specific flood volume for sub-catchments (J, K, L, M, N, O, P, R, S) were calculated and





zero-one variables were established to develop logistic regression model. For computed values of specific flood volume ($\kappa \geq$
$13~m^3 \cdot ha^{-1}$) the variable value was denoted as 1, while in the opposite case it was 0 (Siekmann and Pinekamp, 2011).

**3.3.5. Developing a logistic regression model – simulator specific flood volume (module 5)**

Logistic regression model (LRM) is a tool used for classification. This model has been already applied for modelling

stormwater flooding (Szeląg et al., 2020), identifying stormwater flooding from manholes (Jato – Espino et al., 2018) and the
technical condition of sewage systems (Salman and Salem, 2012). The logistic regression model is described by the following
equation:
$$p_m = \frac{\exp(\alpha_0 + \alpha_1 \cdot x_1 + \alpha_2 \cdot x_2 + \alpha_3 \cdot x_3 + \cdots + \alpha_i \cdot x_i)}{1 + \exp(\alpha_0 + \alpha_1 \cdot x_1 + \alpha_2 \cdot x_2 + \alpha_3 \cdot x_3 + \cdots + \alpha_i \cdot x_i)} = \frac{\exp(X)}{1 + \exp(X)} = \frac{exp(X_{rain} + X_{SWMM} + X_{Catchm})}{1 + exp(X_{rain} + X_{SWMM} + X_{Catchm})}$$    (2)
where $p_m$ – probability of a specific flood volume (understood as the need to corrective actions the stormwater network); $\alpha_0$ –
absolute term; $\alpha_1$, $\alpha_2$, $\alpha_3$, $\alpha_i$ – values of coefficients estimated with the maximum likelihood method, X – vector describing the
linear combination of the independent variables; $X_{rain}$/ $X_{SWMM}$/ $X_{Catchm}$ – vector describing linear combination of statistically
significant:
(a) rainfall characteristics ($X_{rain} = \sum_{s=1}^{t} \alpha_s \cdot x_s$),
(b) SWMM parameters ($X_{SWMM} = \sum_{k=1}^{m} \alpha_k \cdot x_k$),
(c) catchment characteristics, and stormwater network characteristics confidence level – 0.05 ($X_{Catchm} = \sum_{p=1}^{r} \alpha_p \cdot x_p$); $x_i$ –
independent variables describing rainfall characteristics, e.g., rainfall depth, its duration, and the parameters calibrated in the
SWMM, catchment characteristics (permeability, terrain retention, density of stormwater network, length, slope, retention in
stormwater channels etc.).
Independent variables in the logit model were calculated using the forward stepwise algorithm, recommended for the creation
of such models. At the same time, it also ensures the elimination of correlated independent variables (Harrell 2001). The
estimation of the coefficients $\alpha_i$ in Equation (4) and thus the determination of the logistic regression model involved two stages:
learning (80%) and testing (20%). Optimisation of the $p_m$ threshold, equations for determining measures of fit between
computational results and measurements was provided in Supplementary Information (Section 2). A validation of the obtained
logistic regression was additionally performed using the SWMM model for 35 rainfall events (catchment characteristics and
topological data were analysed for separated sub-catchments J, O, S within ±20%), in order to assess the extent of applicability
of the obtained model.

**3.3.6. Sensitivity analysis (module 6)**

According to literature data (Morio, 2011), despite simplifications, local sensitivity analysis is widely applied at the

calibration stage and while analysing the hydrodynamic catchment models. In our study, the sensitivity coefficient was
calculated from the equation (Petersen et al. 2012):


$$S_{xi} = \frac{\partial p_m}{\partial x_i} \cdot \frac{x_i}{p_m}$$
(3)

Where, knowing that $\frac{\partial p_m}{\partial x_i} = \beta_i \cdot p_m \cdot (1 - p_m)$, after transformations, the following formula was obtained (Fatone et al. 2021):
$$S_{xi} = \beta_i \cdot x_i \cdot (1 - p_m)$$
(4)

Value of the $S_{xi}$ was calculated for calibrated SWMM parameters (Table S1), at the same time analysing the impact of rainfall
duration ($t_r = 30 - 90$ min) for rainfall depth $P_t = 10$ mm (representative value for analysing stormwater network functioning
according to DWA – A 118, corresponding to a heavy rainfall event). For the above assumptions, $S_{xi}$ was determined for
different catchment characteristics, which at the same time helped to evaluate the interactions between rainfall data and the
parameter SWMM.

The probability of the specific flood volume ($p_m$) was computed using the logistic regression model for the sub –

catchment characteristics defined in Table 2 and SWMM parameters established during calibration (Szeląg et al., 2016) for
maximum convection rainfall intensity for $t_r = 30$ min and $P_t = 9.62$ mm for Kielce (Section 3 at Supplementary Information).
The calculations of Szeląg et al. (2022) proved that in the urban catchment in question there is a hydraulic overload of the
stormwater system due to convective rainfall. At the same time, the sensitivity coefficients for calibrated SWMM model
parameters were calculated. On this basis the spatial variability of $S_{xi}$ for the sub-basins was determined.

**3.3.7. Application of the logit model to analyse stormwater operating and catchment management (module 8)**

If the stormwater network ceases to function properly and the threshold value of $p_m$ is exceeded, some possible

improvements were suggested, including: (a) increasing the retention depth of impervious areas, i.e. an increase of $d_{imp}$ from
2.50 mm to 3.50 mm, and at the same time raising the Manning roughness coefficient from $n_{imp} = 0.025$ m$^{-1/3} \cdot$s to $n_{imp} = 0.035$
m$^{-1/3} \cdot$s, (b) an increase of hydraulic capacity by reducing the Manning roughness coefficient for stormwater channels from $n_{sew}$
$= 0.018$ m$^{-1/3} \cdot$s to $n_{sew} = 0.012$ m$^{-1/3} \cdot$s. In addition, the possible change of spatial development of urban catchment area was
taken into consideration. Finally, combinations of the above-mentioned computation variants were analysed. When the values
of independent variables (catchment characteristics) adopted for the calculations exceeded the lower/upper (e.g., for Imp =
0.32 - 0.41) limit of applicability of the determined logit model, the simulation results were verified with the mechanistic
model. The verification procedure consisted of three steps:
a) computation of the probability of specific flood volume for rainfall with durations in the range of $t_r = 30 - 90$ min to assess
stormwater network operating,
b) simulation with a calibrated hydrodynamic model for rainfall data as in step (a),
c) comparison of computation results obtained in steps (a), (b); in the event of a of good fit, i.e., proper identification of specific
flood volume, the results obtained from the logit model can be accepted. Three specific corrective variants have been defined
as presented in Table S2.

**3.3.8. Probability of stormwater network failure (module 9)**





The probability of failure (Sun et al., 2012; Karamouz et al., 2013) was used to analyze the performance of the sewage
network in a rainfall event. In the calculations, a failure was defined as an episode (assumed rainfall data, catchment
characteristics, sewer network, SWMM parameters described by *a posteriori distribution* - GLUE results discussed in Section
3.3.3) in which $\kappa \geq 13 m^3 \cdot ha^{-1}$ ($p_m \geq p_{m,cr}$) is exceeded. However, the probability of failure was calculated from the equation:
$$p_F = \frac{\sum_{j=1}^{N} Z_j}{N}, \ where: Z_j = \begin{cases} 1; \ p_m \geq p_{m,cr} \\ 0; \ p_m < p_{m,cr} \end{cases} \tag{5}$$

where: $p_m$ – probability of specific flood volume (exceedance of this value indicates a failure), $p_F$ – probability of the stormwater
network failure in the event of rainfall, $Z_j$ – function describing stormwater network operation, for $Z_j = 1$ – drainage system requires
modernisation; otherwise, i.e. $Z_j = 0$ – modernisation is not necessary.
Based on Equation (5) for the assumed characteristics (rainfall, catchment, drainage network), the operating conditions of the
stormwater network were determined. Hence, an algorithm is given to calculate the performance improvement of a sewer network
in the context of failure probability ($p_F$) reduction. The above effect was obtained by introducing thresholds of maximum permissible
values of Manning roughness coefficients of sewers $n_{sew(m)}$. It was assumed that if the value of nsew (the value from the *a posteriori*
*distribution*) exceeds the maximum permissible value - $n_{sew(m)}$ and determines the occurrence of failure ($Z_j = 1$) and the need to
modernize the sewers, it should be corrected in such a way that $p_m < p_{m,cr}$. The above calculations were reduced to the following
steps:
a) *a posteriori distribution* of calibrated SWMM model parameters (N = 5000 samples),
b) computation of probability of specific flood volume for N items and establishment of failure probability,
c) computation of the Manning roughness coefficient for channels when $p_m > p_{m,cr}$ from the following formula:
$$n_{sew} = \frac{1}{\alpha_{nsew}} \cdot \left[ ln\left(\frac{p_{m,cr}}{1-p_{m,cr}}\right) - \left(\sum_{k=1}^{m-1} \alpha_k \cdot x_k\right) - X_{rain} - X_{Catchm} \right] \tag{6}$$

where: k = 1, 2, 3, …, m – calibrated SWMM model parameters; k = 1, 2, 3, …, m; $\alpha_{nsew}$ – estimated coefficient in logistic regression
model for the Manning roughness coefficient for channels (derivation of the Equation 6 was presented in the Supplementary
Information – Section 4),
d) establishment of empirical distribution describing the $n_{sew}$ values calculated from Equation (6),
e) computation of $n_{sew}$ values from Equation (8) for $n_{sew(un)} \leq n_{sew(m)}$ (where: $n_{sew(un)}$ – Manning roughness coefficients of channels
computed in step (a), $n_{sew(m)}$ – maximal boundary (threshold) value of Manning roughness coefficient for channels), when $n_{sew(un)} \geq$
$n_{sew(m)}$ to $n_{sew} = n_{sew(un)}$,
f) computation of probability of specific flood volume and probability of failure ($p_F$),
g) determination of empirical distribution (CDF) for $n_{sew}$,
h) steps e – g are repeated r = 1, 2, 3, .., z – for different values of $n_{sew,max}$ and median values of $n_{sew(0.5)} = f(n_{sew(m)}, r)$ are denoted based
on empirical distributions,
i) steps a–h are conducted for different catchment characteristics,



j) graph $p_F = f(n_{sew(0.5)})$ is drawn.

**4. Results**
**4.1. Uncertainty analysis – GLUE (module 3)**
Based on SWMM simulation results including uncertainty of calibrated parameters (Table S1), the likelihood functions
were determined (Kiczko et al., 2018). For the observational events (30 May 2010 and 8 July 2011) used to identify the SWMM
parameters, it was found that 96% of the measurement points included the calculated confidence interval. For the validation sets,
90% of the observation points fall within the bands for the 15 September 2010 event and 70% for 30 July 2010 (Figure S1). The
results of the likelihood function calculations for the calibrated SWMM model parameters are given in Figures S2 – S3 in
Supplementary Information.

**4.2. Simulations of stormwater network operation with regard to uncertainty (module 4)**
The results of variation of specific flood volume for the separated sub-catchments has been presented in Figure 3. Based on
the obtained curves it was stated that the uncertainty of SWMM parameters influenced the simulation results, which was confirmed
by the great variability of the 1% and 99% percentile values for each sub-catchment.

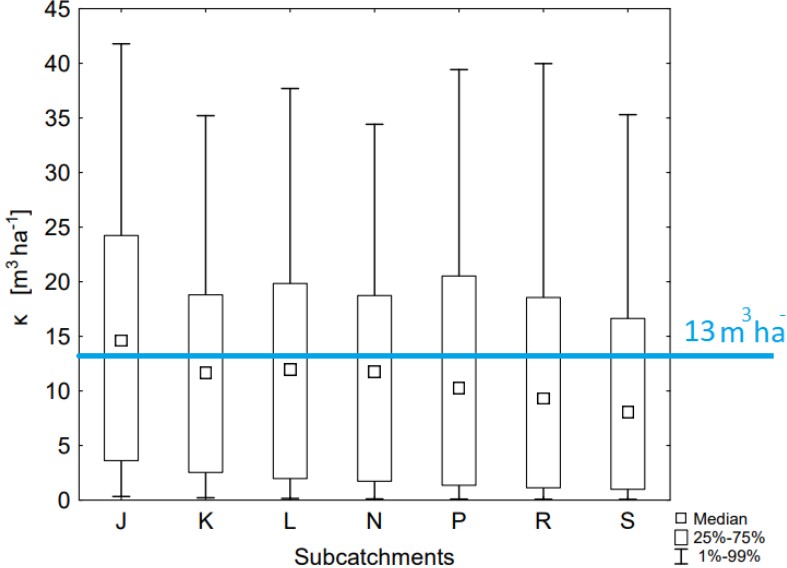

**Figure. 3. Variability of specific flood volume for sub-catchments.**

The median values, enabled to identify that the largest specific flood volume was for sub-catchment J (14.90 $m^3 \cdot ha^{-1}$), and 8.29 $m^3 \cdot ha^{-1}$
$^1$ for the sub-catchment S (Figure 3). The simulation results for the 1% percentiles showed that for adopted rainfall events ($P_t > 5.0mm$
and $t_r < 150$ min) stormwater flooding occurred in all sub-catchments. It was demonstrated that problems with operating of the





stormwater network are present in each sub-catchment, since the calculated values of percentiles (75%, 99%) are higher than 13
m³·ha⁻¹. This indicates that the stormwater network requires modernisation.

**4.3. Determination of the logistic regression model (module 5)**

A LRM was built based on the operational simulation of the stormwater network. The model can be used to identify specific

flood volume and for decision-making regarding corrective actions of the stormwater system. The relationship from Equation (2)
was described by the following linear combination:

$$X_{rain} = 4.05 \cdot P_{tot} - 0.18 \cdot t_r - 54.15 \tag{7}$$

$$X_{SWMM} = 0.23 \cdot \alpha - 79.40 \cdot n_{imp} + 6.23 \cdot \beta + 0.33 \cdot \gamma + 234.12 \cdot n_{sew} \tag{8}$$

$$X_{Catchm} = 76.72 \cdot Imp + 40.77 \cdot Impd - 0.01 \cdot Vk - 1967.04 \cdot Gk - 1169.00 \cdot Gkd - 20.33 \cdot Jkp \tag{9}$$

For other independent variables (Table S2) the determined coefficients were statistically insignificant in prediction confidence band
0.05. Standard deviations of the coefficients estimated from the logit model and the test probabilities are presented in Table S2. The
best fit of the computed results to the measurement data was obtained for $p_{m,cr}$ = 0.75. For the test data set (20%) the following values
were obtained: SPEC = 95.24%, SENS = 84.62% and Acc = 87.87%.

For the determined independent variables (Equation 7, 8), calculations were performed with the LRM and SWMM model

(for 35 rainfall events, $P_t \geq 5$ mm and $t_r \leq 120$ min) assuming values of catchment characteristics and topological data within ±20%
in the separated sub-catchments. The simulation variants analysed and calculation results are given in Table S4 – S11. The results
obtained confirm that the determined LRM model can be applied in a wider range than shown in Table 1. The maximum difference
in the number of events when κ > 13 m³·ha⁻¹ by the ML model and SWMM for Imp = 0.26 - 0.50, Impd = 0.32 - 0.66, Gk = 0.0068
- 0.011 m³·ha⁻¹, Gkd = 0.0009 – 0.0013 m³·ha⁻¹ does not exceed 4 episodes, which confirms the usefulness of the model.

**4.4. Sensitivity analyses (module 6)**

For rainfall depth $P_{tot}$ = 10 mm and duration $t_t$ = 30 – 90 min, the sensitivity coefficients for the SWMM model were

determined, based on Equation (4). For calculation of $S_{xi}$ the values established during calibration were adopted (Kiczko et al., 2018).
The computation results for two parameters of the SWMM model ($\beta$ and $n_{imp}$,) are presented in Figure 4. These two parameters
appeared to have the most significant impact on specific flood volume and, at the same time, they present a vastly different impact
on the dynamics of changes regarding $S_{xi} = f(t_r, Imp, Impd, Vk, Jkp)$; the calculation results for the other SWMM model parameters
are given in Figures S4–S8 (Supplementary Information).

The Figure 4 and Figures S4 – S8 indicated that for the adopted values of $t_r$ and Imp, Impd, Vk, Jkp, the highest values of

$S_{xi}$ was obtained for correction coefficient percentage of impervious areas ($\beta$), Manning roughness coefficient for sewer
channels ($n_{sew}$) and Manning roughness coefficient for impervious areas ($n_{imp}$). Retention depth of impervious areas ($d_{imp}$) had
the lowest impact on the results of specific flood volume. An increase of rainfall duration results in higher values of $S_\beta$, $S_{nimp}$



(Figure 4). The lowest sensitivity coefficients were obtained for $t_r = 30$ min while the highest for $t_r = 90$ min. An increase of
Imp, Impd results in a decrease of $S_\beta$ and $S_{nimp}$ sensitivity coefficients.

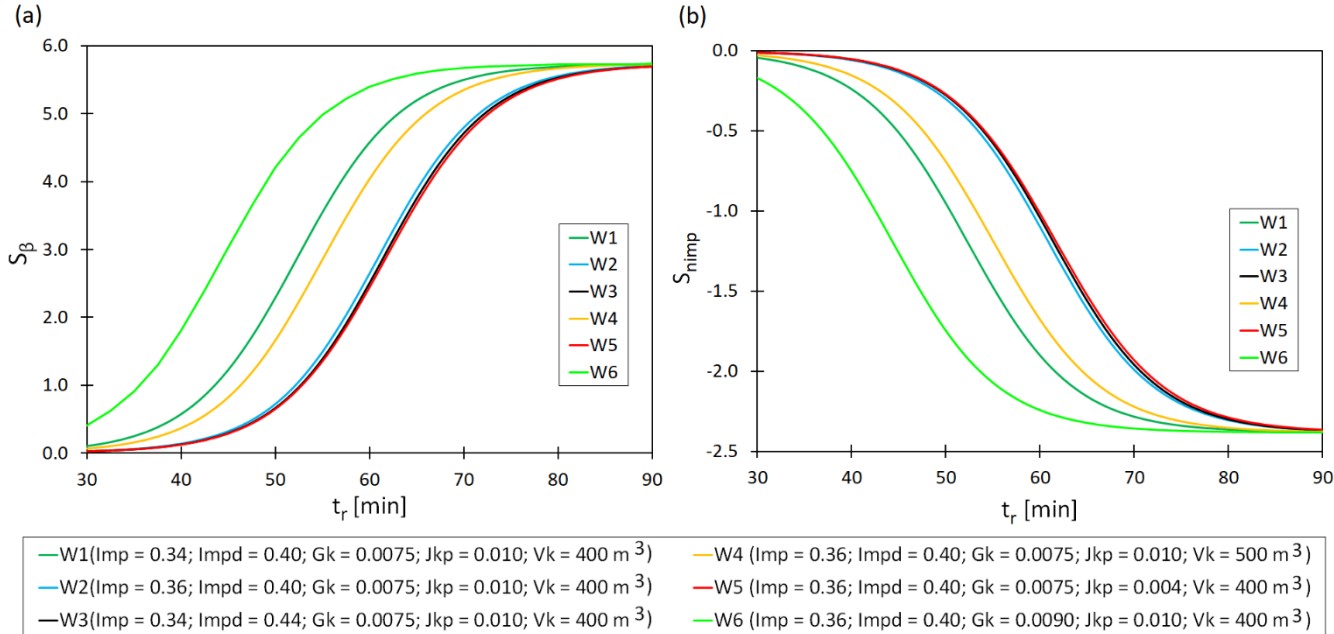


**Figure 4. The impact of rainfall duration ($t_r$) and catchment characteristics (Imp, Impd, Vk, Jkp) on sensitivity coefficients:**
**(a) $S_\beta$, (b) $S_{nimp}$.**

For instance, an increase of Imp from 0.34 to 0.36 results in a decrease of $S_\beta$ from 1.23 to 0.28; identical values were obtained
for Impd (Figure 4). Moreover, an increase of Vk, Jkp, Gk leads to an increase of $S_\beta$ and $S_{nimp}$ sensitivity coefficients. Among
analysed catchment characteristics, density of stormwater network (Gk) had the highest impact on sensitivity coefficients,
while longitudinal slope of canal (Jkp) was of the lowest significance, which is confirmed by variability of obtained curves for
subsequent SWMM parameters (Figure 4).For example, when Vk increased from $400m^3$ to 500 $m^3$, $S_\beta$ increased from 0.29 to
0.82. Additionally, a 10% growth of $S_\beta$ was observed due to a change of Jkp = 0.004 to Jkp = 0.010. Finally, when Gk increased
from 0.0075 to 0.009 $S_\beta$ also increased from 0.29 to 3.03 (Figure 4).
**4.6. Implementation of logit model to analyse the operating of the stormwater network and catchment management**
**(module 7 & 8)**

Due to the fact that in the analysed stormwater network an exceedance of specific flood volume was observed,

possible improvements to the network were considered in terms of correcting catchment imperviousness (Imp) as well as
enhanced terrain retention and channel capacity. The results of $p_m$ computations are presented in Figure 5, while Figure 6
shows $S_\beta$ for variants I, II and III for sub-catchments. Simulation results for the sensitivity coefficients of other SWMM model
parameters (Table S1) and the probability of specific flood volumes are presented in Figures. S9–S17.





430  A decrease of Imp by 10% in sub-catchment J has negligible impact on $p_m$ value, while in sub-catchment S it results

431 in the decrease of specific flood volume probability by 10% (Figure 5a, 5b). It was found that decrease of catchment

432 imperviousness (variant I) leads to improvement of stormwater system operation (Figure 5). The greatest reduction in volume

433 flooding was obtained for variant III, when $p_m$ values decreased by 2% and 36% for sub-catchments J and S (Figure 5d).

435 **Figure 5. Probability of specific flood volume in sub-catchments for: (a) present state ($p_0$) and for (b) I, (c) II, (d) III**
436            **corrective actions variants.**

437 Based on the $p_m$ values in catchments J, M, N, S for corrective action variant III, it was found that, despite the increase in

438 retention depth, channel capacity and reduction in imperviousness of the catchments, there was hydraulic overloading ($\kappa > 13$

439 $m^3 \cdot ha^{-1}$) in the sub-catchments. This indicates the need for further changes to both the catchment and the stormwater network

440 than was assumed. For variants I, III the Imp values for the sub-catchment are below the applicability range of the logit model,

441 so mechanistic model simulations were performed to verify the results (Table S4). The results of the model calculations confirm

442 their high agreement; out of 72 cases, identical results were obtained in 68 cases. The calculations performed (variant I, II, III)





for the sub-catchment showed a greater influence of changes in terrain retention and channel capacity on the sensitivity
coefficients than the probability of specific flood volume (Fig. 6). For catchments J, S, a 10% decrease in Imp (variant I)
increased $S_\beta$ by 7.55 times and 17.50 times (Fig. 6a, 6d). For variant II (increasing catchment retention), sensitivity coefficients
were found to be higher than 51% (catchment S) and 59% (catchment J) compared to variant I, and the highest $S_\beta$ was obtained
in variant III. The $S_\beta$ values for sub-catchment S are higher than in catchment J by 20.7 times, 19.3 times and 14.7 times for
variants I, II and III, respectively. These results provide relevant information for planning retention infrastructure that reduces
outflow.

**Figure 6. Sensitivity coefficient ($S_\beta$) in sub-catchments for: (a) present state (0) and for (b) I, (c) II, (d) III**
**corrective action variants.**

**4.7. Probability of failure (module 9)**

Based on SWMM model parameters determined via the MCM method (Table S1), probability of failure ($p_F$) was

computed for convection rainfall in Kielce with a duration time of $t_r$=30 min and $P_{tot}$= 9.61 mm. The following threshold values





of $n_{sew(m)}$ were adopted for calculations: $n_{sew(m)} = 0.015 - 0.045$ m$^{-1/3}$·s, coupled with three variants of catchment characteristics:
Imp = 0.36 and Impd = 0.40; Imp = 0.35 and Impd = 0.40; Imp = 0.35 and Impd = 0.42. The impact of canal retention (Vk =
750, 850, 950 m$^3$); density of stormwater network (Gk = 0.0075, 0.0080, 0.0085 m·ha$^{-1}$; Gkd = 0.005, 0.006, 0.007 m·ha$^{-1}$) in
upper and lower part of the catchment on probability of failure ($p_F$) was also analysed. The Manning roughness coefficients of
the channels ($n_{sew}$) for the analysed variants were presented as empirical distribution (CDF). In Figure 7a, 8a the results for
Imp = 0.36, Impd = 0.40 and Vk = 750, 850, 950 m$^3$ are presented, while other variants are shown in Figures S18, S19.

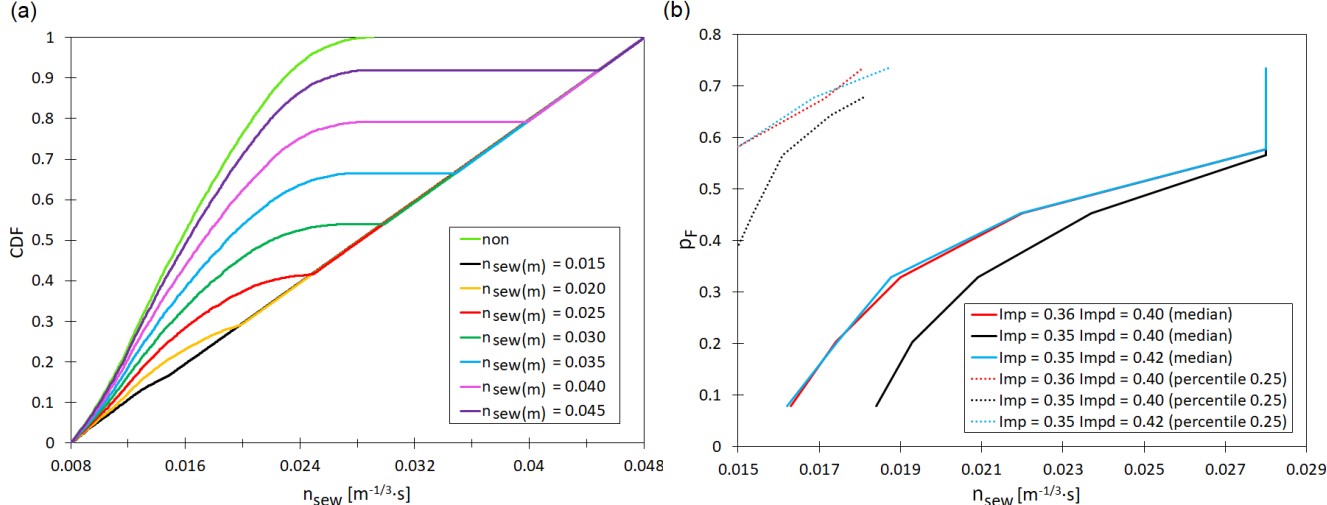


**Figure 7. (a) Empirical distributions of threshold values of Manning roughness coefficients of channel ($n_{sew}$). (b) Impact of Manning roughness coefficient of channel on failure probability ($p_F$) in relation to Imp, Impd.**

Figure 7b presents the impact of $n_{sew}=f(n_{sew(m)})$ for percentiles 0.25 and 0.50 (based on the curves in Figures 7b, 8b, 8c, 8d,
S25, S26 the values of the respective percentiles for the analysed $n_{sew(m)}$) on the probability of failure ($p_F$). Assuming that
Manning roughness coefficients – $n_{sew(un)}$ determined by MC simulation which exceeds the threshold triggers the corrective
actions of sewer pipes resulting in reduction of roughness below $n_{sew(m)}$ following the condition in which the stormwater
network functions $p_m = f(X_{rain}, X_{SWMM}, X_{Ctchm}) > 0.75$ for an independent rainfall event, it was found out, that an
appropriate decrease of percentiles (0.25 and 0.50 - median) leads to improved network operation and to a lower failure
probability (Figures. 7a, 7b). It was observed that the change of percentile 0.50 for $n_{sew}$ for a sample from MC simulation leads
to a decrease from 0.028 m$^{-1/3}$·s to 0.021 m$^{-1/3}$·s (as a result of correction $n_{sew(un)} < n_{sew(m)}$) and to improved stormwater network
operation understood as a lower probability of failure (decrease of $p_F$ from 0.68 to 0.42 for Imp = 0.36 and Impd = 0.40). These
results confirm the significance of catchment characteristics (Imp, Impd) for the operability of a stormwater network. For Impd
= 0.40, the reduction in catchment impervious area (Imp) from 0.36 to 0.35, at percentile $n_{sew} = 0.019$ m$^{-1/3}$·s results in a
decrease in failure probability from $p_F = 0.42$ to $p_F = 0.33$ (Figure 7b).

Great impact of channel retention (Vk) and density of stormwater network in the upper and lower part of a catchment

(Gkd and Gk, respectively) on probability of failure $p_F$ were indicated (Figure 8). For $n_{sew} < 0.0215$ m$^{-1/3}$·s $p_F$ reached higher



values (max. 0.41) than for $V_k = 850$ m$^3$ and $V_k = 950$ m$^3$. The highest failure probability ($p_F = 0.80$) was obtained for $V_k =$
750 m$^3$ ($n_{sew} = 0.031$ m$^{-1/3}$·s), while the lowest $p_F = 0.65$ was obtained for $V_k = 950$ m$^3$ (Figure 8b).




**Figure 8. (a) Empirical distributions of threshold values of Manning roughness coefficients of channels ($n_{sew}$) for**
**$V_k = 950$m$^3$. Impact of Manning roughness coefficient for channel on failure probability ($p_F$) in relation to: (b) $V_k$ –**
**canal retention, (c) $G_k$ - length of stormwater channel per impervious area in a catchment (m·ha$^{-1}$), (d) $G_{kd}$ - length of**
**a channel per impervious area below closing cross-section (m ha$^{-1}$).**


Furthermore, the highest probability of failure $p_F = 0.79$ was obtained for $G_k = 0.0075$ m·ha$^{-1}$ ($n_{sew} = 0.031$ m$^{-1/3}$·s), while the
lowest for $G_k = 0.0085$ m·ha$^{-1}$   ($n_{sew} = 0.0276$ m$^{-1/3}$·s) (Figure 8c). It was established that for $n_{sew} < 0.023$ m$^{-1/3}$·s computed
values of $p_F$ for $G_k = 0.0075$ m·ha$^{-1}$ and $G_k = 0.0080$ m·ha$^{-1}$ are higher than 0.41. Moreover, the highest failure probability $p_F$
for $n_{sew} = 0.035$ m$^{-1/3}$·s was equal to 0.82 for $G_{kd} = 0.005$ m·ha$^{-1}$, while for $G_{kd} = 0.007$ m·ha$^{-1}$ it was 0.73 (Figure 8d).





**5. Discussion**

Developing and calibrating mathematical models to simulate stormwater network operation under hydraulic overloads

is one of the latest areas of research. In comparison to the models used so far (Li and Willems, 2019; Thorndahl 2009), the
logistic regression model proposed in this study includes SWMM model parameters describing catchment retention and, at the
same time, the characteristics of the catchment and stormwater network (Table 4).

**Table. 4. Comparison of developed model for identification of specific flood volume to literature data**

| Study | Criteria | M | I | R | C | S | P |
|---|---|---|---|---|---|---|---|
| Duncan et al. (2011) | occurrence of flooding | ✓ | ● | ✓ | ✓ | ✓ | ● |
| Jato - Espino et al. (2018) | occurrence of flooding | ✓ | ✓ | ✓ | ✓ | ✓ | ● |
| Jato - Espino et al. (2019) | occurrence of flooding | ✓ | ● | ✓ | ✓ | ✓ | ● |
| Li and Willems (2020) | occurrence flooding | ✓ | ✓ | ✓ | ✓ | ✓ | ● |
| Szeląg et al. (2021) | volume | ✓ | ✓ | ✓ | ✓ | ✓ | ✓ |
| Szeląg et al. (2022a) | occurrence of flooding | ● | ● | ✓ | ✓ | ✓ | ✓ |
| Szeląg et al. (2022b) | specific flood volume | ✓ | ✓ | ✓ | ● | ● | ✓ |
| Thorndahl et al. (2008) | volume | ✓ | ✓ | ✓ | ● | ✓ | ✓ |
| Verbovski et al. (2022) | volume | ✓ | ✓ | ✓ | ● | ● | ● |
| Fu et al. (2011) | volume | ● | ● | ✓ | ✓ | ✓ | ✓ |
| Chen et al. (2020) | volume | ● | ● | ✓ | ✓ | ✓ | ✓ |
| Fraga et al. (2016) | volume | ● | ● | ✓ | ✓ | ✓ | ✓ |
| this study | specific flood volume | ✓ | ✓ | ✓ | ✓ | ✓ | ✓ |


where: M (method); the models were divided into two groups: mechanistic (·) and statistical model (∨); R (rainfall); C
(catchment); S (sewer); P (calibration parameter); I (interpretation model, based on estimated factors the impact of analysed
factors on stormwater flooding can be determined).

Apart from the model developed in this study, the above-mentioned factors are only included in MCM, which have a form of
differential equations. Therefore, they require a large number of simulations in order to determine the impact of selected
variables on computation results of specific flood volume. Free from such drawbacks are statistical models (Table S4) that
take the form of empirical relationships. For models developed with neural networks, there is a need of performing additional
analyses (Ke et al, 2020; Yang et al., 2020). Jato – Espino et al. (2018, 2019) and Li and Willems (2020) analysed stormwater
flooding from manholes based on catchment characteristics and stormwater network characteristics (Table 4). Szeląg et al.
(2022) confirmed their results and developed a model for identification of stormwater flooding in a catchment, but not
considered catchment retention. In this context, the approaches cited above were insufficient to analyse the impact of different
types of pavement (for example roof, road, parking etc.) on sewage flooding. Fu et al. (2011), Thorndahl et al. (2009), Szeląg
et al. (2022b) analysed the uncertainty of the identified parameters, which allowed, for example, to correct for impervious area





retention, roughness coefficient without being able to correct for catchment imperviousness, which limited the use of the
models in catchment management. The approach proposed in this study is a combination of these two solutions, which provides
a tool which can be successfully implemented to manage other catchments.
The results of this study confirmed the major significance and huge interaction between catchment characteristics and
SWMM model parameters. This fact can be further compared by several references (Li and Willems, 2020; Jato – Espino et
al., 2019; Zhuo et al., 2019) presenting comparisons of flooding simulations in urban catchments. This analysis indicated that
an impervious area in a catchment (Imp, Impd) leads to the increase of flooding; reverse dependency was obtained by Jato –
Espino et al. (2018) when modelling flooding from manholes. Increase in channel volume above the closing cross-section of
a catchment (Vk) and its longitudinal slope (Jkp) results in the decrease of flooding, that was confirmed for Espoo catchment
in Finland (Jato – Espino et al. 2018). The increase of unit impervious area per the length of main stormwater interceptor (Gk,
Gkd) results in smaller volume of stormwater flooding. This is due to the relationship that the longer the channel, the greater
the number of manholes. Huang et al. (2018) based on observations conducted in a complex stormwater system indicated the
impact of catchment location and hydrological conditions on the peak flow of flooding. Yao et al. (2019) obtained similar
results after computations with a MCM for catchments in Beijing and in Dresden (Reyes – Silva et al. 2020).
Calculation results obtained in this study confirmed relevant impact of rainfall data, catchment characteristics, and
stormwater network characteristics on sensitivity coefficients – relationships between SWMM parameters and specific flood
volume. For rainfall data and catchment characteristics (assumed as constant) it was proved that correction coefficient of
impervious area ($\beta$) and the Manning roughness coefficient for channels ($n_{sew}$) have the greatest impact on specific flood
volume. The results of this computations were consistent with Thorndahl et al. (2009), who simulated flooding from a single
manhole in the Frejlev catchment (Belgium), based on rainfall data and calibrated parameters of a MCM. These findings were
confirmed by calculations Fu et al. (2012) and Prodanovic et al. (2022) respectively for catchments of 400 ha and 8 ha. Szeląg
et al. (2021, 2022b) based on simulations with MCM including uncertainty of SWMM parameters proved the key impact of
Manning roughness coefficient of sewers on specific flood volume (for rainfall event $t_r$ = 30 min and $P_t$ = 15.25 mm). Fraga
et al. (2016) used GLUE+ GSA method for a road catchment and indicated the impact of rainfall data (rainfall duration, depth,
temporal distribution) on sensitivity analysis results. It was confirmed in computations of stormwater flooding using logit
model (Szeląg et al. 2022) and specific flood volume calculations with SWMM model (Freni et al. 2012). Xing et al. (2021)
used MCM to determine characteristics of spatial development and stormwater characteristics in Chongqing catchment (China)
on the depth of stormwater flooding. The aforementioned research studies indicate the impact of rainfall data, catchment
characteristics, and stormwater network characteristics on sensitivity of hydrodynamic simulation model for stormwater
flooding.
The sensitivity analysis development proposed in this study enabled its application for catchments with different
characteristics, which is an improvement compared to previously applied, more specified approaches (Cristiano et al. 2019;
Fatone et al., 2021). Differences in probability of occurrence/sensitivity coefficients indicate the influence of catchments
downstream on conditions in the catchment above. The variation in sensitivity coefficients does not account for local conditions



within the side channels. Due to the creation of successive sub-catchments by combining them, the conditions of the sewer system in its area are averaged out, making the interpretation of the results difficult. Using the developed tool, catchment management may become difficult when there is a particularly hydraulically overloaded area within the catchment, which impacts neighbouring sub-catchments.

As in the case to the sensitivity analysis, in this study the extension of the sewer system failure assessment has been adapted to enable the implementation for a random catchment (for the sewer system without pump stations). Calculations outputs showed the influence of the catchment and sewage network characteristics on the failure probability. The introduction of the maximum allowable value of the Manning roughness coefficient for the sewer channel, enabled to model the improvement of the operating conditions of the sewage network under uncertainty. A similar approach was used in the study of Fu et al. (2012) by limiting to probabilistic rainfall characteristics (Del Giudice, et al. 2013) and using a MCM to simulate the drainage system. Fu et al. (2011) modified the above approach by focusing on the impact of uncertainty in the calibrated parameters on flooding; however, it was not possible to analyse retention, channel capacity on system performance.

**6. Conclusions**

In this study a novel simulator of logistic regression extended by advanced risk assessment was developed for modeling stormwater systems operation under uncertainty. The proposed model is an alternative approach to mechanistic models, that can be used at the preliminary stage of analyses related to spatial planning, urban development and expansion etc. This is of major significance since at the preliminary stage, the data set for building catchment models is limited and urgent demand for simulation algorithm to assist decision making is required. Assuming Manning roughness coefficients – $n_{sew(un)}$ estimations that exceed the threshold triggers corrective actions of sewer pipes resulting in a reduction of roughness below $n_{sew(m)}$ following the condition of proper functioning of the stormwater network ($p_m > p_{mcr}$). Appropriate decrease of percentiles (0.25 and 0.50 - median) led to improved network operation and to a lower failure probability requirement.

In the adopted hydrodynamic model (based LRM), the impact of rainfall data, catchment characteristics (impervious areas in the downstream and upstream) and stormwater network characteristics (the length of channel per unit impervious area, channel slope and volume) as well as SWMM parameters (roughness coefficient for sewer channel, correction coefficient for percentage impervious area Manning roughness coefficients for impervious area) were included simultaneously. The obtained simulations results show the strong interaction between the above-listed parameters. This is extremely relevant in the context of models calibration that can be applied to analyse stormwater network operation and to support the decision-making process (management of stormwater in an urban catchment). Since the proposed solution analyses the spatial distribution of sensitivity coefficients, it is possible to identify the most vulnerable areas inside a catchment that require specific attention while identifying SWMM model parameters, which could also be taken into account when locating measuring facilities.





**7 Appendices**
**Appendix A: List of Symbols**
Symbols:
$A_{imp}$ – area of impervious surface (ha),
$dH1$ – height difference of the terrain at section above closing cross-section (m),
$dHp$ – height difference at section above closing cross-section (m),
$CDF$ – Cumulative Distribution Function (–),
$d_{imp}$ – retention depth of impervious areas (mm),
$d_{perv}$ – retention depth of pervious areas (mm),
$F$ – catchment surface area (ha),
$Gk$ – length of stormwater channel per impervious area in a catchment (m·ha$^{-1}$),
$Gkd$ – length of a channel per impervious area below closing cross-section (m·ha$^{-1}$),
$GLUE$ - Generalized Likelihood Uncertainty Estimation,
$Hst$ – the height of a manhole at closing cross-section (m),
$Imp$ – impervious area,
$Impd$ – impervious area of a catchment of downstream area,
$J$ – average rainfall intensity (l·(s·ha)$^{-1}$),
$Jkp$ – channel slope above closing cross-section of a catchment
$K$ – total number of sewer manholes (–),
$Lk$ – length of channel above closing cross-section of a catchment (m),
$L(Q/\theta)$ – likelihood function,
$n_{imp}$ – Manning roughness coefficient for impervious areas (m$^{-1/3}$·s),
$n_{perv}$ – Manning roughness coefficient for pervious areas (m$^{-1/3}$·s),
$n_{sew}$ – Manning roughness coefficients of sewer channels (m$^{-1/3}$·s),
$Q_z$ – denote $z$-th value from the times series of observed and computed discharges (m$^3$·s$^{-1}$),
$P_t$ – maximum depth of rainfall (mm),
$p$ – cumulative distribution function (CDF),
$p_m$ – probability of specific flood volume,
$P(\theta)$ – stands for *a priori* parameter distribution,
$R.t.$ – height difference of the channel (m),
$S_{xi}$ – sensitivity coefficient,
$x_i$ – independent variables,
$SWMM$ – Storm Water Management Model,





$t_r$ – duration of rainfall (min),
V () – variance,
Vk – volume of stormwater channel (m$^3$),
Vkd – total retention of a catchment.
Vkp – volume of the channel above the closing cross-section of a catchment (m$^3$),
Vrd – catchment retention above the closing cross-section (m$^3$),
$V_{t(i)}$ – floodings volume from $i$ - th sewer manhole (where: $i$ = 1, 2, 3, …, k) (m$^3$),
$W$ – width of the runoff path in a subcatchment (m),
$\alpha$ – Coefficient for flow path width (–),
$\beta$ – Correction coefficient for percentage of impervious areas (–),
$\gamma$ – Correction coefficient for subcatchment slope (–),
$\varepsilon$- a scaling factor for the variance of model residua, used to adjust the width of the confidence intervals,
$\kappa$ – specific flood volume (m$^3 \cdot$ha$^{-1}$),

**Code availability:** The authors announce that there is no problem sharing the used model and codes upon request to the
corresponding author.

**Data availability:** The authors confirm that data supporting the findings of this study are available from the corresponding
author upon request.

**Author contribution:** Conceptualization: Szeląg, Methodology: Fatone, Szeląg, Kiczko; Formal analysis and investigation:
Szeląg, Kiczko, Stachura, Wałek; Writing - original draft preparation: Szeląg, Kowal, McGarity, Wojciechowska, Wałek,
Fatone, Caradot; Writing - review and editing: Kowal, Wojciechowska, McGarity, Fatone, Caradot; Supervision: Szeląg,
Kowal, McGarity, Wojciechowska, Caradot.
**Competing interests:** The authors declare that they have no conflicts of interest.

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
