# Peer review of "An advanced tool integrating failure and sensitivity analysis to novel modeling for stormwater flooding volume"

_Hydrology and Earth System Sciences, 2023_

## Author Comment (AC1)

**Reviewer1.** The preprint states that: "The developed simulator to determine a specific flood volume represents an alternative approach to the SWMM model that, unlike current approaches, is calibratable with limited topological data availability, therefore generates a lower cost due to the less amount and specificity of data required." and this is very interesting and important, but no information is given regarding the accuracy loss, or not, or the error in relation to the traditional swmm model, please insert information.

In the revised manuscript, the following modifications were made:

"For the determined independent variables (Equation 7, 8), calculations were performed with the LRM and SWMM model (for 35 rainfall events,  $P_t \ge 5$  mm and  $t_r \le 120$  min) assuming values of catchment characteristics and topological data within  $\pm 0.2$  in the separated sub-catchments. The results of the validation of the developed model for the identification of the specific flood volume are given in Tables S5 - S11. In the range of NF(SWMM) = (0 - 6), the relative difference in the number of episodes when  $\kappa \ge 13$  m3·ha-1 did not exceed 20%, and for NF(SWMM) = <6, 19> was 15 - 33%.

**Figure XX**. Comparison of LRM and SWMM simulation results of the number of episodes when the specific flood volume was greater than  $13m^3 \cdot ha^{-1}$  (where:  $N_{F(SWMM)}$  – prediction of SWMM,  $N_{F(LRM)}$  – prediction of LRM; \* - minimum, maximum values of the catchment characteristics, topology of the stormwater network in Table 1; yellow - the upper limit of the model, blue - the lower limit of the model).

The maximum difference between LRM and SWMM simulations ( $N_{F(SWMM)} - N_{F(LRM)} = 4$ ) was obtained for Imp = 0.49, Impd = 0.66, Gk = 0.011 m·ha-1, Vk = 1500 m3, which corresponds to the extreme values of the catchment characteristics, the topology of the sewer network. Verification results showed that the maximum difference in the number of events when  $\kappa > 13$  m3·ha-1 by the ML model and SWMM for Imp = 0.26 - 0.50, Impd = 0.32 - 0.66, Gk = 0.0068 - 0.011 m3·ha-1, Gkd = 0.0009 - 0.0013 m3·ha-1 did not exceed 4 episodes (Figure XX). The calculations performed confirm the high fitting of the calculations with measurements of the number of episodes when the specific flood volume exceeds 13 m3·ha-1." (L 400 – 414).

,,A validation of the obtained logistic regression was additionally performed using the SWMM model for 35 rainfall events (catchment characteristics and topological data were analysed for separated sub-catchments J, O, S within  $\pm 20\%$ ), in order to assess the extent of applicability of the obtained model. In this study, 35 independent rainfall events were assumed for model validation, for which  $P_t = 6.0 - 15.0$  mm and  $t_r = 30 - 120$  min. For validation of the LRM model, catchments J, O, S were selected, in which characteristics of catchment (Imp, Impd) and topology network (Gk, Gkd, Jkp) were varied in the interaction scheme. At the variant generation step, combinations of two inputs were used to verify model, then values of which were changed in a three-point scheme -0.2/0/+0.2. The calculations included the following steps:

a) selection of two input data (x1, x2) to change; the values of the other parameters were taken as the mean of the data according to Table 1,

b) determination of combinations x1, x2 for verification calculations such that:  $1.2 \cdot x1 - 1.2 \cdot x1$ ,  $1.2 \cdot x1 - x2$ ,  $1.2 \cdot x1 - 0.8 \cdot x2$ ; x1 -  $1.2 \cdot x2$ , x1 - x2, x1 -  $0.8 \cdot x2$ ;  $0.8 \cdot x2 - 1.2 \cdot x1$ ,  $0.8 \cdot x2 - x1$ ,  $0.8 \cdot x2 - 0.8 \cdot x1$ ; all combinations of catchment and sewer network characteristics were analysed in this study, resulting in a total of 135 verification variants for 3 sub-catchments (135-35-3 = 14175 simulations),

c) modification of sub-catchment characteristics according to point b)

d) calculation with a logit model and SWMM of the value of the specific flood volume."

(L61 – 70), Support Information.

---

## Author Comment (AC2)

**Reviewer 2.** The topic of the article is interesting and stimulating. Here you can find my comments:

**1. OVERALL: Please, enlarge and re-arrange font sizes to guide the reader properly in all sections. All figures must be composed of HD images. It is mandatory to improve the scientific quality of the whole manuscript.**

The quality of the figures has been improved throughout the manuscript, and the font size for the main chapters has been increased to 12, and the subchapters to 10. In addition, graphic programs were used to improve the quality of the maps (S9 - S17); a sample drawing of S9 is provided.

In the revised manuscript, the following modifications were made:

Current version: Highlight After modification: Highlight (L36) Current version: Introduction After modification: Introduction (L41) Current version: Case study After modification: Case study (L100) Current version: Methodology After modification: Methodology (L100) Current version: Results After modification: Results (L365)

Figure. 1. Study catchment area (Wałek, 2019).

---

## Author Comment (AC3)

**Reviewer1.** The preprint states that: "The developed simulator to determine a specific flood volume represents an alternative approach to the SWMM model that, unlike current approaches, is calibratable with limited topological data availability, therefore generates a lower cost due to the less amount and specificity of data required." and this is very interesting and important, but no information is given regarding the accuracy loss, or not, or the error in relation to the traditional swmm model, please insert information.

In the revised manuscript, the following modifications were made:

,,*For the determined independent variables (Equation 7, 8), calculations were performed with the LRM and SWMM model (for 35 rainfall events, $P_t \geq 5$ mm and $t_r \leq 120$ min) assuming values of catchment characteristics and topological data within $\pm 0.2$ in the separated sub-catchments. The results of the validation of the developed model for the identification* of the specific flood volume are given in Tables S5 - S11. In the range of $N_{F(SWMM)} = (0 - 6)$, the relative difference in the number of episodes when $\kappa \geq 13$ m$^3 \cdot$ha$^{-1}$ did not exceed 20%, and for $N_{F(SWMM)} = <6, 19>$ was 15 - 33%.

[Figure]

**Figure XX**. Comparison of LRM and SWMM simulation results of the number of episodes when the specific flood volume was greater than 13m$^3 \cdot$ha$^{-1}$ (where: $N_{F(SWMM)}$ – prediction of SWMM, $N_{F(LRM)}$ – prediction of LRM; * - minimum, maximum values of the catchment characteristics, topology of the stormwater network in Table 1; yellow - the upper limit of the model, blue - the lower limit of the model).

The maximum difference between LRM and SWMM simulations ($N_{F(SWMM)}$ - $N_{F(LRM)}$ = 4) was obtained for Imp = 0.49, Impd = 0.66, Gk = 0.011 m·ha$^{-1}$, Vk = 1500 m$^3$, which corresponds to the extreme values of the catchment characteristics, the topology of the sewer network. Verification results showed that the maximum difference in the number of events when $\kappa$ > 13 m$^3$·ha$^{-1}$ by the ML model and SWMM for Imp = 0.26 - 0.50, Impd = 0.32 - 0.66, Gk = 0.0068 - 0.011 m$^3$·ha$^{-1}$, Gkd = 0.0009 – 0.0013 m$^3$·ha$^{-1}$ did not exceed 4 episodes (Figure XX). The calculations performed confirm the high fitting of the calculations with measurements of the number of episodes when the specific flood volume exceeds 13 m$^3$·ha$^{-1}$." (L 400 – 414).

 In this study, 35 independent rainfall events were assumed for model validation, for which $P_t$ = 6.0 - 15.0 mm and $t_r$ = 30 - 120 min. For validation of the LRM model, catchments J, O, S were selected, in which characteristics of catchment (Imp, Impd) and topology network (Gk, Gkd, Jkp) were varied in the interaction scheme. At the variant generation step, combinations of two inputs were used to verify model, then values of which were changed in a three-point scheme -0.2/0/+0.2. The calculations included the following steps:

a) selection of two input data (x1, x2) to change; the values of the other parameters were taken as the mean of the data according to Table 1,

b) determination of combinations x1, x2 for verification calculations such that: 1.2·x1 - 1.2·x1, 1.2·x1 - x2, 1.2·x1 - 0.8·x2; x1 - 1.2·x2, x1 - x2, x1 - 0.8·x2; 0.8·x2 - 1.2·x1, 0.8·x2 - x1, 0.8·x2 - 0.8·x1; all combinations of catchment and sewer network characteristics were analysed in this study, resulting in a total of 135 verification variants for 3 sub-catchments (135-35-3 = 14175 simulations),

c) modification of sub-catchment characteristics according to point b)

d) calculation with a logit model and SWMM of the value of the specific flood volume."
(L61 – 70), Support Information.

---

## Author Comment (AC4)

**Reviewer 2.** **The topic of the article is interesting and stimulating. Here you can find my comments:**

**1. OVERALL: Please, enlarge and re-arrange font sizes to guide the reader properly in all sections. All figures must be composed of HD images. It is mandatory to improve the scientific quality of the whole manuscript.**

The quality of the figures has been improved throughout the manuscript, and the font size for the main chapters has been increased to 12, and the subchapters to 10. In addition, graphic programs were used to improve the quality of the maps (S9 - S17); a sample drawing of S9 is provided.

In the revised manuscript, the following modifications were made:

Current version: **Highlight**

After modification: **Highlight** (L36)

Current version: **Introduction**

After modification: **Introduction** (L41)

Current version: **Case study**

After modification: **Case study** (L100)

Current version: **Methodology**

After modification: **Methodology** (L100)

Current version: **Results**

After modification: **Results** (L365)

[Figure]

**Figure. 1. Study catchment area (Wałek, 2019).**

[Figure]

**Figure. 1. Study catchment area (Wałek, 2019).** (L104)

current version

[Figure]

**Figure. 3. Variability of specific flood volume for sub-catchments.**

after modification

[Figure]

**Figure. 3. Variability of specific flood volume for sub-catchments.** (L381)

current version

[revised manuscript text omitted]

**3. METHODS: Please, insert a Figure for each sub-section. This could improve considerably the clarity of your manuscript.**

Dear Reviewer, we appreciate the comment. We would like to kindly inform you that the option in which each computational scheme was to be included in a subsection was considered, but it led to a significant lengthening of the manuscript. Thus, we ultimately decided that the individual computational components of the model and its subsequent steps would be presented in the computational algorithm. The assumed approach in our the paper is typical of the field literature, as evidenced by exemplary publications:

1. Jato-Espino, D., Sillanpää, N., Andrés Doménech, I., Rodríguez-Hernández, J. Flood risk assessment in urban catchments using multiple regression analysis. J. Water Resour. Plann. Manag. 144, 04017085 (2018). https://doi.org/10.1061/(ASCE) WR.1943-5452.0000874.

2. Mayra, R., Fu, G., and Butler, D., Yuan, Z., Cook, L. Global Resilience Analysis of Combined Sewer Systems Under Continuous Hydrologic Simulation. J Environ Manage 344, 118607 (2023). http://doi.org/10.2139/ssrn.4416250

3. Jato-Espino, D., Indacoechea-Vega, I., Gáspár, L., Castro – Fresno, D. Decision support model for the selection of asphalt wearing courses in highly trafficked roads. Soft Comput 22, 7407–7421 (2018). https://doi.org/10.1007/s00500-018-3136-7

4. Jato-Espino, D., Martín-Rodríguez, Á., Martínez-Corral, A., Sañudo-Fontaneda, L. A. Multi-expert multi-criteria decision analysis model to support the conservation of paramount elements in industrial facilities. Herit Sci 10, 68 (2022). https://doi.org/10.1186/s40494-022-00712-7

5. Casal-Campos, A., Sadr, S.M.K., Fu, G., Butler, D. Reliable, resilient and sustainable urban drainage systems: an analysis of robustness under deep uncertainty. Environ. Sci. Technol. 52, 9008 (2018). https://doi.org/10.1021/acs. est.8b01193. –9021